# Sexual and reproductive health research capacity strengthening programs in low- and middle-income countries: A scoping review

Julie M. Buser[1]*, Anna Grace Auma[2], Ella August[3,4], Gurpreet K. Rana[5], Rachel Gray[1], Faelan E. Jacobson-Davies[1,6], Tesfaye H. Tufa[7], Tamrat Endale[1], Madeleine Mukeshimana[8], Yolanda R. Smith[1,6]

1 Center for International Reproductive Health Training (CIRHT), University of Michigan, Ann Arbor, Michigan, United States of America, 2 Department of Nursing and Midwifery, Lira University, Lira, Uganda, 3 Department of Epidemiology, University of Michigan School of Public Health, Ann Arbor, Michigan, United States of America, 4 PREPSS (Pre-Publication Support Service), University of Michigan, Ann Arbor, Michigan, United States of America, 5 Taubman Health Sciences Library, University of Michigan, Ann Arbor, Michigan, United States of America, 6 Department of Obstetrics and Gynecology, University of Michigan, Ann Arbor, Michigan, United States of America, 7 St. Paul Institute for Reproductive Health and Rights (SPIRHR), Addis Ababa, Ethiopia, 8 College of Medicine and Health Sciences, University of Rwanda, Kigali, Rwanda

* jbuser@umich.edu

## Abstract

Sexual and reproductive health (SRH) research capacity strengthening (RCS) programs in low- and middle-income countries (LMICs) are needed to foster the discovery of context-specific solutions to improve patient outcomes and population health. There remains a limited understanding of SRH research strengthening programs to raise skill sets, publications, and infrastructure and ultimately influence health policy and patient outcomes in LMICs. More information is needed to understand how SRH research is sustained after program completion. To inform efforts to implement programs that strengthen SRH research and foster sustainability, we conducted a scoping review to identify and synthesize strategies used in SRH research strengthening programs in LMICs. A literature search of nine scholarly databases was conducted. We synthesized data extracted from included articles and presented results highlighting the format, duration, and topics covered of program interventions to strengthen SRH research in LMICs. We organized information about primary outcomes into themes and summarized how SRH research capacity was sustained after program completion. Twenty-four articles were included in the scoping review. The articles generally focused on outcomes within the themes of advocacy/capacity, education, policy, project life cycle, and writing/publication. Few articles reported metrics or other evidence of long-term program sustainability of SRH RCS projects in LMICs. Results from this scoping review can be used to strengthen SRH research programs in LMICs. More energy must be directed toward correcting power imbalances in capacity strengthening initiatives. To address additional gaps, future directions for research should include an exploration of SRH research mentorship, the cost of SRH RCS interventions, and how to foster institutional support.

**Data Availability Statement:** The data supporting their findings can be found in the supplemental appendix mentioned in the text.

**Funding:** An anonymous grant to YS at the Center for International Reproductive Health Training at the University of Michigan provided funding. The funders had no role in study design, data collection and analysis, decision to publish, or preparation of the manuscript.

**Competing interests:** The authors have declared that no competing interests exist.

## Introduction

Research capacity strengthening (RCS) programs in low- and middle-income countries (LMICs) are needed to foster the discovery of context-specific solutions to improve patient outcomes and population health. In the context of sexual and reproductive health (SRH), the World Health Organization (WHO) Human Reproduction Program defines RCS as a process of individual and institutional development that leads to higher skill level and greater ability to perform valuable research that is linked to improved operations of SRH programs [1, 2]. RCS would necessitate understanding the current strengths and goals of the collaboration. In practice, RCS approaches are broad and include supporting partnerships, improving the capacity of individual researchers through training courses, and developing leadership skills [3].

A comprehensive approach is needed to strengthen independent researcher capacity sustainably and improve SRH in LMICs. Researchers need the skills to build on local evidenced-based research outcomes to effectively reduce maternal mortality while improving SRH research. Over the past several decades, numerous RCS reviews have been conducted, and various frameworks have been proposed outlining approaches for capacity strengthening in LMICs [4–14]; however, none, to our knowledge, focused explicitly on SRH. There remains a limited understanding of reproductive health research strengthening programs and their outcomes. More information is also needed to understand how SRH research is sustained after program completion. Strengthening our understanding in this realm will help to inform health policy in LMICs.

Researchers and institutions with strong research programs can ultimately influence health system strategy, drive policy changes, and shift power between LMICs and high-income countries (HICs) [15–17]. Because a large share of health research funding in LMICs comes from HICs, financial control and decision-making power are often constrained in these settings [18]. As a result, health interventions are often dictated by those outside the country where they take place. This power disparity is further reflected in research publications from these settings [19]. Resident researchers most familiar with the local context and values are often excluded from the research [20]. Even when participating in research, they are often not given appropriate credit [21]. The sponsor, researchers, and pertinent stakeholders, including community boards and host-country authorities, should engage in dialogue and negotiation to establish and attain precise capacity-strengthening goals [22]. These parties should collaborate to enhance research capacity within the country's health system, ensuring its sustainability for continued knowledge generation [22]. It's crucial for local principal investigators to be actively engaged in the research endeavor [22]. Greater research capacity will result in more research carried out by local researchers, resulting in local knowledge production created within the priorities and values of residents [19, 23].

We conducted a scoping review of the literature to identify and synthesize strategies used in SRH research strengthening programs in LMICs. The scoping review explores the existing literature and aims to describe the format, duration, topics covered, and outcomes of program interventions to strengthen SRH research in LMICs. The review also explores how SRH research capacity is sustained after program completion in LMICs and identifies gaps in the literature about how SRH research programs influence patient outcomes and health policy. The overarching goal is to use the literature to inform efforts to successfully implement current and future programs to strengthen SRH research programs while fostering long-term sustainability and acknowledging the power imbalances that hold progress back in LMICs.

## Materials and methods

The scoping review followed the Preferred Reporting Items for Systematic Reviews and Meta-analyses Extension for Scoping Review (PRISMA-SCr) guidelines [24]. A comprehensive

search of the literature published between January 2011 and August 2023 was conducted by a health sciences informationist (GKR). Discrete searches were conducted in nine databases: MEDLINE via Ovid, Embase via Elsevier, Scopus, CINAHL via Ebsco, Web of Science, Psy-cINFO via Ebsco, Women's Studies International via Ebsco, CABI Global Health and Global Index Medicus. No restrictions on language were applied. The original literature search was implemented in November 2021. A search update was conducted in August 2023. The time-frame of 2011–2023 was chosen to capture the most recent and relevant literature in the field. This period ensured coverage of current research trends, methodologies, and findings while managing the scope of the review. Focusing on the last decade also aligns with the need to explore contemporary issues and developments in SRH research strengthening programs in LMICs.

Search strategies were constructed by combining search terms representing three search concepts: (1) sexual and reproductive health, (2) research strengthening or capacity building, and (3) low- and middle-income countries (LMIC) or low-resourced regions. The third concept was represented by variations on the Cochrane Effective Practice and Organisation of Care (EPOC) LMIC search filter (v.4) [25] in all databases except for Embase, CABI Global Health, and Global Index Medicus. The University of North Carolina's Developing Country / Low-Middle Income Searches filter (2019 update) [26] was used in Embase. No LMIC search filters were used in CABI Global Health or Global Index Medicus. The World Bank list of country income status was used as a criterion to identify a country as an LMIC [27].

After removing duplicate articles in EndNote using the Bramer method [28] and removal of identified retracted articles in Zotero, a total of 2315 articles were exported to Rayyan (https://www.rayyan.ai/) for title and abstract screening. Additional literature was also identi-fied through hand searching of cited references of articles selected for full-text screening.

## Study selection

**Inclusion and exclusion criteria.** Regarding inclusion criteria, articles needed to describe an SRH RCS program, be conducted in an LMIC, and be published from 2011–2023 in peer-reviewed journals. The WHO definition of health research was used, encompassing the follow-ing areas of activity: measuring the health problem; understanding its cause(s); elaborating solutions; translating the solutions or evidence into policy, practice and products; and evaluat-ing the effectiveness of solutions [29]. The included articles needed to describe programs aimed at allied and public health professionals. Qualitative, quantitative, case study, and mixed methods studies were included. Table 1 shows the inclusion and exclusion criteria for the review. Reporting a program evaluation was not part of the inclusion criteria.

**Table 1. Inclusion and exclusion criteria for this scoping review paper including literature published 2011–2023.**

| Inclusion Criteria: | Exclusion Criteria: |
|---|---|
| Describes a SRH[1] research capacity strengthening program | Not a SRH research capacity strengthening program |
| Program was conducted in LMIC[2] | Conducted in a high-income country |
| Papers published in the years ranging from 2011 to 2023. | Papers published before 2011. |
| Paper describes programs aimed at allied or public health professionals | Not aimed at allied or public health professionals |
| Qualitative, quantitative, case study, or mixed methods studies | Reviews, commentaries, editorials, or conference proceedings |
| Peer-reviewed publications | Publications not peer-reviewed |

[1]sexual and reproductive health

[2]low- or middle-income country

**Screening process.** At least two authors (JB, EA, RG, FJD, TE, YS) reviewed and coded each article. The team reconciled disagreements through screening and discussion with at least one additional coder. If a coder was uncertain whether an abstract met inclusion criteria, the article was put forth for full-text screening. During the full-text screening, the same procedures for review were followed as in the title and abstract screening. Articles found through hand-searching reference lists of all identified articles for full-text screening were also screened for inclusion. S1 Appendix provides a data extraction form with bibliographic information for all full-text articles (n = 57) screened for inclusion.

**Data extraction.** Relevant information for included articles was directly extracted using a structured form that included information on the lead author, title, and year of publication (S1 Appendix). The goal of the SRH research capacity strengthening program along with the country where it was located were included as was a summary of the sample size, population, and study design. We report the SRH research intervention frequency and format along with learning activities. Text describing reported intervention outcomes and key findings was extracted. Information about the sustainability of the program and limitations identified by researchers was also extracted by authors (JB, RG, FJD).

**Data analysis.** We synthesized data extracted from included articles and presented results to highlight the format, duration, and topics covered of program interventions to strengthen SRH research in LMICs. We organized information about primary outcomes into themes and summarized how SRH research capacity was sustained after program completion.

## Results

### Program characteristics

A total of twenty-four articles were included in the scoping review. The PRISMA flow diagram is shown in Fig 1. Fifty-one articles found through hand searching of reference lists were screened, but none were deemed appropriate for inclusion. While the search was inclusive of non-English articles, none were included in the final review because they did not meet inclusion criteria.

Out of the twenty-four included articles, eleven focused on countries within Africa [30–40]. The authors also focused on countries within Asia [41–43] and South America [44–46]. One article focused on Oceania (Papua New Guinea) [47], and two articles focused broadly on LMICs without indicating specific countries [48, 49]. Finally, one article focused on countries within the Middle East and North Africa (MENA) region [50].

More articles were published between 2018–2023 (n = 14) than from 2011–2017 (n = 10); this upward trend reflects a recent focus on SRH RCS in LMICs. In most publications, researchers in HICs led RCS programs. A majority (n = 10) of the studies identified themselves as case studies [30, 33–35, 40, 44, 47, 48, 53, 54]. The next most frequent classification was studies defined as mixed methods, wherein qualitative and quantitative data are presented [38, 43–45].

### Principles

In terms of program design and approach, 12 studies offered in-person training on SRH topics to participants in LMICs. Face-to-face training formats included modules or workshops [11, 30, 33, 35, 38, 39, 42, 43, 50], networking opportunities at health colloquiums and conferences that offered a venue for engaging with policymakers, methodological training and definition of priority research topics and research methodology [33, 36], and lectures [46]. Six programs developed a training partnership to guide researchers [30, 34, 41, 45, 48, 54]. Some of the selected articles described programs that supported research projects through direct funding

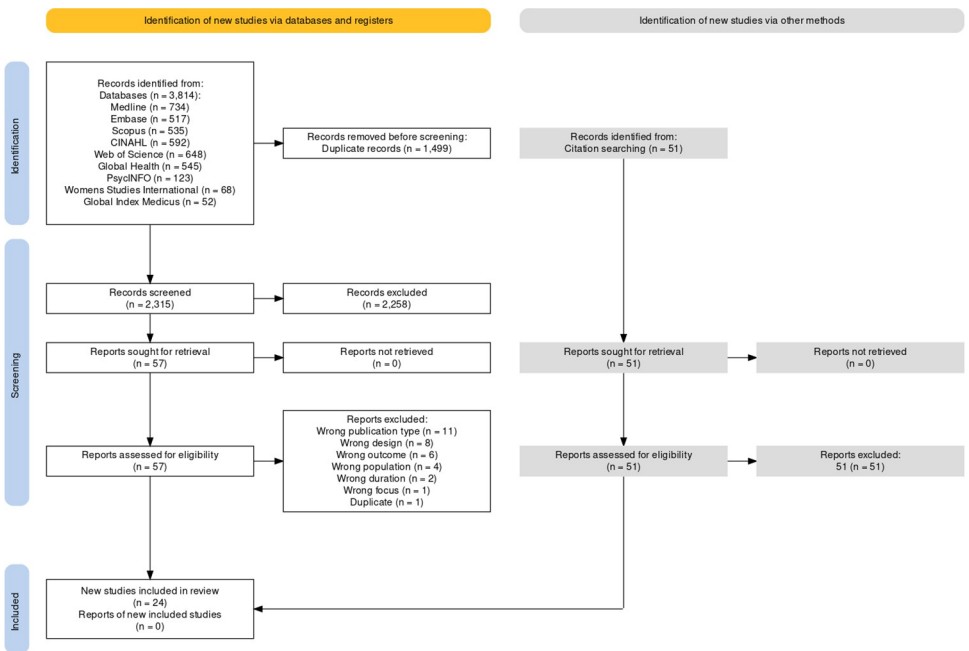

**Fig 1. Summary of the number of article exclusions at each stage of this scoping review of interventions in low- and middle-income countries in 2011–2023.** This diagram is based on the Preferred Reporting Items for Systematic reviews and Meta-Analyses guidelines [24, 51, 52].

[32, 44], and one project studied researchers who had actively participated in a previously funded study [47]. One study examined the impact of a single meeting involving researchers, communication specialists, and donors [40]. Another developed a manual of research procedures [31]. Lastly, one study sought to understand the trends in reproductive health in humanitarian settings and organizational changes over time using a questionnaire [49].

Program size ranged from large groups with over two hundred participants [38, 55] to smaller groups of approximately eight participants [50]. Larger group sizes were possible because of online teaching formats. Participants also had varied professional backgrounds within allied and public health. Five studies included fellows and other early career professionals [31, 36, 47, 48, 55]. Two studies focused on participants from diverse backgrounds, such as nurses, midwives, doctors, community health workers, and public health technicians [30, 55]. One study specified working with a mix of early and advanced career professionals [54].

When considering the time frame, most of the included studies lasted less than a year, ranging from one day to eight months [35, 38, 40, 47, 49, 53–55]. Six studies lasted between one and three years [31–33, 36, 39, 42]. The longest studies lasted between four and seven years [30, 34, 41, 44, 46].

Looking at topics covered in SRH RCS programs, most studies focused on bringing research into medical practice [38, 39, 42, 45, 46, 49, 53, 55]. Six studies were interested in increasing the capacity to conduct research and collect data [30, 32, 34, 41, 43, 47], while others studied how the findings of SRH research are analyzed and disseminated [33, 44, 54]. The remaining studies focused on strengthening research and policy links [31, 35, 36, 40, 50] and proposing a framework for co-production of capacity strengthening health research [30].

Using the data extraction form (S1 Appendix), articles reporting similar primary outcomes were grouped and summarized into six themes by two of the authors. The articles generally focused on outcomes within the themes of advocacy/capacity, education, policy, project life

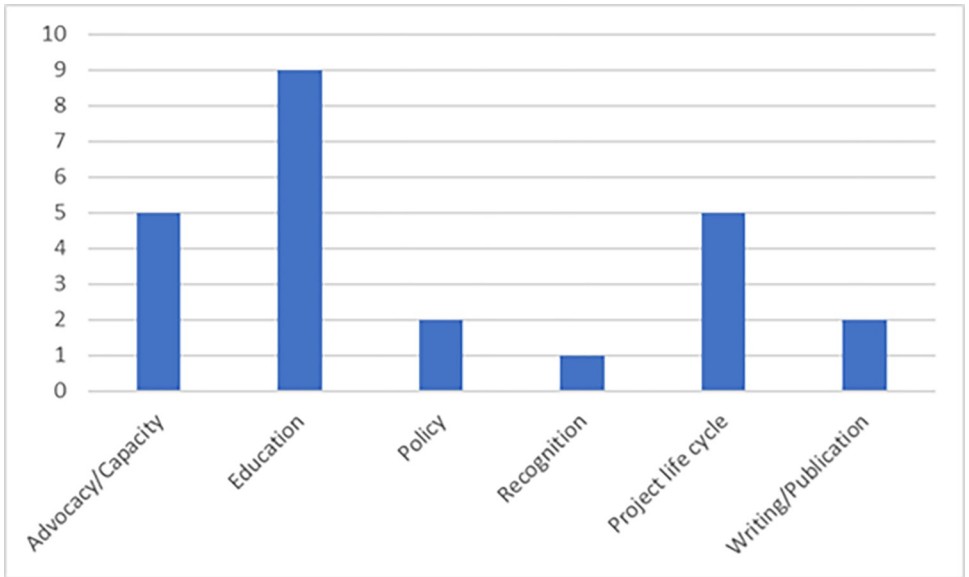

**Fig 2. Articles' primary outcomes for this scoping review paper including literature published 2011–2023.**

cycle, and writing/publication (Fig 2). Most articles described outcomes within the realm of education, including incorporating research components into the curriculum in nursing, midwifery, and medical schools [34, 42, 43, 46, 55]. For those describing education outcomes, self-reported SRH research competencies increased after a three-day course of interactive lectures [46], as did post-intervention test scores after a professional development program with interactive modules [43]. We acknowledge the limitations and bias of self-reporting, and the results of studies utilizing self-reporting are likely non-generalizable. However, studies utilizing self-reporting techniques can still provide a general understanding of participants' perspectives on the efficacy of training. Midwifery educators reported increased confidence while linking international standards and WHO competencies to midwifery school education curriculum revisions [4], and Millimouno [55] reported that blended learning using Sexual and Reproductive Health Services Management (eSSR) helped strengthen capacity in monitoring and data management. For Elmusharaf and colleagues [34], SRH education was expanded and recognized by the Ministry of Health in Sudan, while country-level initiatives, including the inauguration of a research unit, were established.

The next most popular outcome themes were advocacy for SRH policy changes and increased funding for RCS programs [32, 39, 45, 49, 53] and strengthening capacity for each part of the project life cycle [31, 45, 47, 50]. Authors also mentioned increasing publications [44, 54], policy [40], and recognizing researchers' accomplishments to enhance motivation and productivity [41]. Seven articles also had a secondary outcome focus, three focusing on policy and the need to disseminate research findings to decision-makers [35, 44, 45].

In terms of lessons learned from SRH research interventions, researchers who authored articles often mentioned the importance of long-term engagement [30, 40, 43, 44], networking [31, 40, 41, 44, 54], and regional partnerships [37, 42, 44, 50, 54]. Five studies identified lessons about intensifying efforts to train researchers to engage with policymakers [12, 34–36, 40]. Another lesson learned was the need to sustain momentum in programs [37, 39, 49].

Not all studies that included lessons learned had formal evaluations of their SRH projects. Only 11 of the included articles described how programs were evaluated. Table 2 shows the

**Table 2. Summary of the subset of articles (n = 11) that included an evaluation of the reproductive health research capacity strengthening program described.**

| Lead author, title, year of publication | Evaluation | Methods used | Indicators | Outcomes | Conclusions |
|---|---|---|---|---|---|
| Elmusharaf, From local to global: a qualitative review of the multi-level impact of a multi-country health research capacity development partnership on maternal health in Sudan, 2016 | Illustrate the long-term effects of a multi-country (8 countries) global partnership for health systems research capacity development (Connecting health Research in Africa and Ireland Consortium—ChRAIC) in relation to its contribution to capacity strengthening, public advocacy and policy influence at different levels and its practical achievements in Sudan in addressing access to maternal health services. | The authors (all members of the global partnership) reflect on the project in one of its' partner countries, Sudan, over its' five year duration. This reflection is supported by specific project data collected over the period of the project (2008–2014). The data collected included: (i) 6 monthly and annual donor reports; (ii) a mid-term internal and end of project independent evaluation of the entire project, and; (ii) a Ph.D study conducted by a member of the Sudanese research team. | The ChRAIC project in Sudan achieved the deliverables set out at the beginning of the project. These included a national knowledge synthesis report on Sudan's health system; identification of country level health systems research priorities; research capacity assessment and skills training, and; the training and graduation of a Sudanese team member with a Ph.D. | Mechanisms established in Sudan to facilitate these achievements included the adoption of culturally sensitive and locally specific research and capacity strengthening methods at district level; the signing of a Memorandum of Understanding at country level between the Ministry of Health, research and academic institutions in Sudan, and; the establishment of country level initiatives and a research unit. The latter being recognized globally through awards and membership in global health forums. | The 'network of action' approach adopted to partnership formation facilitated the benefits gained, but that adopting such an approach is not sufficient. More local and contextual factors influenced the extent of the benefits and the sustainability of the network. |
| Ezenduka, Evaluating a capacity development intervention in health economics among producers and users of evidence in Nigeria: a case study in Getting Research Into Policy and Practice (GRIPP) in Anambra State, 2022 | Evaluate the impact of the workshop training on selected stakeholders on the use of health economics evidence to inform health policy and practice in Anambra state. | Pre-post test approach and group exercises were used to assess the knowledge and impact of the training exercises on the participants regarding the use of health economics evidence. | The participants were administered 15-item test questions before and after the training with the same set of questions concerning the knowledge and principles of health economics and its use in healthcare decision making/priority setting. Group exercises were conducted for the participants to reflect the practice setting of their services. | Pretest average scores varied from 39.7% to 60.5% while posttest scores varied from 47.6% to 65.7%, showing big differences in individual scores among participants, between the producers and users of evidence both prior to and after the training. The significant differences between the test scores indicated success in increasing the knowledge of participants on the use of health economics evidence. | Findings underscore the need for significant and regular upgrade of participants' knowledge on the use of health economic evidence to inform decision making in the state's healthcare system. |
| Kabra, Research capacity strengthening for sexual and reproductive health: a case study from Latin America, 2017 | Review efforts to build research capacity in Latin America by studying and analyzing the 5-year history of institutional development support to an institution in Paraguay. | During the first 5 years of the LID grant (2009–2014), based on gaps identified by the institution, training support was provided on data analysis (quantitative/qualitative), multivariate analysis, scientific writing, dissemination of research results, and library management. | Following the training, technical support from HRP and continuous mentoring by CENEP, CEPEP conducted several research projects. Reorganization and expansion of CEPEP's library. Shift towards publishing in national and international scientific peer reviewed–journals, developing policy briefs, participating and organizing congresses and seminars. Increased networking and visibility. | 1. Greater support is needed from developmental partners and governments to strengthen research capacity in LMICs to improve SRH. 2. Long term support to institutions to attain a certain degree of capacity to conduct quality research in SRH and to sustain the gains made is critical. 3. Increase collaboration between mature and emerging research centres for further strengthening the research environment. 4. Networking and visibility of local institutions will help garner support, increasing recognition and leading to sustainability. | The authors call for greater support from and collaborative efforts of developmental partners and governments to strengthen research capacity in low and middle-income countries to improve sexual and reproductive health. |

*(Continued)*

**Table 2.** (Continued)

| Lead author, title, year of publication | Evaluation | Methods used | Indicators | Outcomes | Conclusions |
|---|---|---|---|---|---|
| Koso-Thomas, The Global Network for Women's and Children's Health Research: A model of capacity-building research, 2015 | Conduct research focused on several high-need areas, such as preventing life-threatening obstetric complications, improving birth weight and infant growth, and improving childbirth practices to reduce mortality. | Review the results of the Global Network's research, the impact on policy and practice, and highlights the capacity-building efforts and collaborations developed since its inception. The assessment and retention of skills among the health workers was evaluated for several studies. | Scientists from developing countries, together with peers in the USA, lead research teams that identify and address population needs through randomized clinical trials and other research studies. | Global Network projects develop and test cost-effective, sustainable interventions for pregnant women and newborns and provide guidance for national policy and for the practice of evidence-based medicine. | Some skill-sets are more reliably replicated than others; however, the capacity of the research teams clearly has grown overall. Ultimately, the continued involvement of a relatively stable number of clusters for such an extended time has been effective, as demonstrated by the ability to conduct large multi-site, multi-country randomized clinical trials with rigor. |
| Millimouno, Evaluation of Three Blended Learning Courses to Strengthen Health Professionals' Capacity in Primary Health Care, Management of Sexual and Reproductive Health Services and Research Methods in Guinea, 2022 | Evaluate the reasons for dropout and abstention, the learners' work behavior following the training, and the impact of the behavior change on the achievements of learners' organizations or services. | Quantitative and qualitative data were collected through an open learning platform, via an electronic questionnaire, during the face-to-face component of the courses (workshops), and at learners' workplaces. | Completion rate, dropout rate, abstention rate were calculated. Overall success rate was the number of learners who passed the course, over the total number of enrollees. | Over the three courses, the completion rates were similar (67–69%) along with 20–29% dropout rates. The majority (87%) of the learners reported applying the knowledge and skills they acquired during the courses through activities such as supervision (22%), service delivery (20%), and training workshops (14%). | Findings showed fair success rates and a positive impact of the training on learners' work behavior and the achievements of their organizations. |
| Millimouno, Outcomes of blended learning for capacity strengthening of health professionals in Guinea, 2021 | Two courses lasting 3 months each (7–8 modules) were developed and implemented: one in Primary Health Care (eSSP) and the other in Sexual and Reproductive Health Services Management (eSSR). Both eSSP and eSSR courses were offered online on the Moodle platform, followed by a face-to-face capacity-building workshop. | A cross-sectional study using a mixed-methods approach was conducted in 2018–19. As outcomes, they described learners' sociodemographic characteristics, course completion and success, and perceptions of the courses and support from the instructors, analyzed the factors associated with learners' successful completion and reported on learners' feedback on their blended learning experience. Quantitative data were analyzed using the STATA 15 software, and qualitative data were analyzed through content analysis. | Overall, 282 health professionals were enrolled for both eSSP and eSSR courses. The completion rate was 69.5% (196/282). The success rate for learners who completed the courses was 80% (156/196), and the overall success rate for enrollees was 55% (156/282). The dropout and abstention rates were 22 and 9%, respectively. On both eSSP and eSSR courses, the success rate of women enrolled was higher than or equal to men's. The success rate of medical doctors enrolled (53% for eSSP and 67% for eSSR) was higher than for other health professionals, in particular nurses (9% for eSSP) and midwives (40% for eSSR). Course type was associated with success (AOR = 1.93; 95% CI = 1.15–3.24). | Most learners strongly agreed that the courses are relevant for targeted health professionals (81 to 150/150), pdf course materials are well-structured and useful (105/150), the content of the modules is relevant, comprehensible, and clear (90/150), self-assessment quizzes are helpful (105/150), summative assessment assignments are relevant (90/150), the course administrators and IT manager were responsive to learners' concerns (90/150), they will recommend the courses to colleagues and friends (120/150). | Two blended courses for capacity strengthening of health professionals were successfully developed and implemented in Guinea. |

(*Continued*)

**Table 2.** (Continued)

| Lead author, title, year of publication | Evaluation | Methods used | Indicators | Outcomes | Conclusions |
|---|---|---|---|---|---|
| Pinto, International participatory research framework: triangulating procedures to build health research capacity in Brazil, 2011 | Showed how advances in Community-based Participatory Research (CBPR) can facilitate participatory research in myriad international settings. Steps and actions were used recursively to build a partnership to study the roles of community health workers (CHWs) in Brazil's Family Health Program (PSF). The research conducted using IPRF focused on HIV prevention, and it included nearly 200 CHWs. | By using procedural triangulation—the combination of specific steps and actions as the basis for the International Participatory Research Framework (IPRF)—their approach can improve the abilities of researchers and practitioners worldwide to systematize the development of research partnerships. The IPRF comprises four recursive steps: (i) contextualizing the host country; (ii) identifying collaborators in the host country; (iii) seeking advice and endorsement from gatekeepers and (iv) matching partners' expertise, needs and interests. | IPRF includes: (A1) becoming familiar with local languages and culture; (A2) sharing power, ideas, influence and resources; (A3) gathering oral and written information about partners; (A4) establishing realistic expectations and (A5) resolving personal and professional differences. | By using the IPRF, their partnership achieved several participatory outcomes: community-defined research aims, capacity for future research and creation of new policies and programs. They engaged CHWs who requested that they study their training needs, and they engaged CHWs' supervisors who used the data collected to modify CHW training. | Data collected from CHWs will form the basis for a grant to test CHW training curricula. Researchers and community partners can now use the IPRF to build partnerships in different international contexts. By triangulating steps and actions, the IPRF advances knowledge about the use of CBPR methods/procedures for international health research. |
| Ropa, Lessons from the first 6 years of an intervention-based field epidemiology training programme in Papua New Guinea, 2013–2018, 2019 | The Papua New Guinea (PNG) National Department of Health and its partners developed a field epidemiology training programme of Papua New Guinea (FETPNG) to strengthen the country's public health workforce. The training programme covers field epidemiology competencies and includes the design, implementation and evaluation of evidence-based interventions by Fellows. | The training programme covers field epidemiology competencies and includes the design, implementation and evaluation of evidence-based interventions by Fellows. Continual adjustments to the FETPNG curriculum have been made in response to feedback from Fellows and faculty. Based on this feedback, more time has now been allocated to teaching practical skills. | From 2013 to 2018, FETPNG graduated 81 field epidemiologists. Most FETPNG graduates (84%) were from provincial or district health departments or organisations. Many of their intervention projects resulted in successful public health outcomes with tangible local impacts. Health challenges addressed included reducing the burden of multi-drug resistant-tuberculosis (TB), increasing immunisation coverage, screening and treating HIV/TB patients, and improving reproductive health outcomes. FETPNG Fellows and graduates have also evaluated disease surveillance systems and investigated disease outbreaks. | The graduates of FETPNG now provide a critical mass of public health practitioners across the country. Their skills in responding to outbreaks and public health emergencies, in collecting, analysing and interpreting data, and in designing, implementing and evaluating public health interventions continues to advance public health in PNG. | A focus on designing and implementing effective public health interventions not only provides useful skills to Fellows but also contributes to real-time, tangible and meaningful improvements in the health of the population. |

*(Continued)*

**Table 2.** (Continued)

| Lead author, title, year of publication | Evaluation | Methods used | Indicators | Outcomes | Conclusions |
|---|---|---|---|---|---|
| Srisaeng, Looking toward 2030: Strengthening midwifery education through regional partnerships, 2019 | The International Confederation of Midwifery's standards of midwifery education and World Health Organization midwifery educator core competencies provided the framework for capacity-building of Lao midwifery educators. | Follow-up evaluation was initiated 4 months after the training to assess application of knowledge and clinical skills into their work as midwifery teachers and preceptors. In-depth interviews with semi-structured questions in accordance with ICM standards and WHO competencies were conducted with the participants, department heads, and co-workers (midwives and obstetricians). | Knowledge gained from this 2-year project (Oct 2015 –Nov 2017) increased the teaching confidence of midwifery educators while linking international standards and competencies to curriculum revision. | Teaching efforts were successful and the newly trained Lao midwife educators sustained their learning and enhanced overall midwifery in the country by providing training to other midwifery educators and students. These newly trained midwife educators gained confidence in their skills through the efforts of the Thai educators and furthermore, voiced the need to extend the training throughout the country. | Partnerships are essential to meeting the sustainable development goals. These regional partnerships may be highly effective in creating sustainable capacity-building projects. |
| Tomatis, Evidence-based medicine training in a resource-poor country, the importance of leveraging personal and institutional relationship, 2011 | Between 2005 and 2009, researchers conducted an annual 3-day course in Perú consisting of interactive lectures and case-based workshops. | Assessed self-reported competence and importance in EBM using a Likert scale (1 = low, 5 = high). | Totally 220 clinicians participated. For phase I (2005–2007), self-reported EBM competence increased from a median of 2 to 3 (P < 0.001) and the perceived importance of EBM did not change (median = 5). For phase II (2008–2009), before the course, 8–72% graded their competence very low (score of 1–2). | After the course, 67–92% of subjects graded their increase in knowledge very high (score of 4–5). The challenges included limited availability of studies relevant to the local reality written in Spanish, participants' limited time and lack of long-term follow-up on practice change. Informal discussion and written evaluation from participants were universally in agreement that more training in EBM is needed. | In an EBM course in a resource-poor country, the baseline self-reported competence and experience on EBM were low, and the course had measurable improvements of self-reported competence, perceived utility and readiness to incorporate EBM into their practices. Similar to developed countries, translational research and building the research capacity in developing countries is critical for translating best available evidence into practice. |
| Vaz, Enhancing the Education and Understanding of Research in Community Health Workers in an Intervention Field Site in South India, 2014 | In a community-based TB study in South India, a Professional Development Programme (PDP) was developed to optimize the quality of the research, and to make involvement in research more meaningful for all members of the research team. The programme started with a set of modules that were considered relevant for a community-based epidemiological study. Subsequently, new modules were developed based on the expressed needs of the study staff. | Pre and post-tests were conducted for all modules. A qualitative assessment was conducted after 3 years. | While initial 'pre-test' knowledge across modules was lower amongst community level field workers than staff of higher cadres, they achieved passing grade scores (>75%) with the programme. Longer term retention of knowledge over 6 months for the basic modules, was in excess of 75% for all staff. | Focus groups discussions revealed that while many in the research team began work with an incomplete understanding of research methods and community health principles, investing time and resources in education beyond protocol training via the PDP realized long-term benefits to the research study and the individual staff, particularly the community level workers. | There is a need to enhance the capabilities of community health workers to be part of a research team in field settings, so as to optimize their performance while utilizing their existing skill sets. |

methods used, indicators, outcomes, and conclusions of only the articles that reported project evaluations.

The research skills covered in RCS projects encompassed a broad range of competencies essential for conducting effective research. These skills included understanding theories of knowledge, research design, conduct, synthesis, interpretation, and utilization [30]. Additionally, the trainings covered stakeholder engagement, communication, and issues management [31], as well as specific methodologies such as concept mapping and health economics [53]. Participants were equipped with practical skills like data analysis, scientific writing, dissemination of research results, and database management [44, 54]. The training programs emphasized experiential learning, peer-to-peer interaction, and evaluation mechanisms to ensure the acquisition and application of these skills in diverse research contexts [30, 39, 42].

Few articles reported metrics or other evidence of long-term program sustainability of SRH research capacity strengthening projects in LMICs. However, most articles proposed methods for how to address sustainability after program completion. Many encouraged researchers to advocate for increased organization expenditure for training about the research process and augmented funding for research grants. Some advised incorporating SRH research into institutional strategic planning. Others recommended partnering with international experts to support researchers through the publication process. Some suggested forming regional partnerships to develop broader training plans or create sustainable capacity-building projects [37, 42, 45, 46]. Elmusharaf [34] proposed understanding a partnership as a 'network of action' to achieve collectively more than as individuals and to have influence beyond the pilot site. Meanwhile, Dossou and colleagues [33] with the Network for Scientific Support in the field of Sexual and Reproductive Health in West and North Africa and Agyepong et al. suggested that supporting LMIC institutions in developing their protocols and focusing on the development of the methodological capacities and the shift of power and responsibilities is vital to long-term sustainability [30].

Regarding power dynamics and locus of decision-making, there were only three examples of south-south RCS collaboration [31, 37, 42]. In South Africa, Baron et al. from Wits Reproductive Health and HIV Institute shared lessons learned from implementing the Good Participatory Practice Guidelines for Biomedical HIV Prevention Trials across multi-party regional research consortia in several sub-Saharan African countries [31]. Meanwhile, Sriseng and Upvall highlighted the importance of their regional capacity building project between Thailand and Laos as essential to meeting sustainable development goal 3 through midwifery education [42]. In all other articles, researchers in HICs or affiliated with the WHO/UN were involved in programs to build the capacity of those in LMICs. None of the articles specifically mentioned who initiated the capacity building program. In four articles, the authors mentioned who decided what the content and process would be for RCS [37, 42, 45, 53].

## Discussion

This scoping review described the format, duration, topics covered, and outcomes of program interventions to strengthen SRH research in LMICs. We also explored whether and how teams addressed the sustainability challenge after SRH research capacity program completion in LMICs. Included articles focused on outcomes within the themes of advocacy/capacity, education, policy, project life cycle, and writing/publication. Program organizers cited lessons learned recognizing the importance of long-term engagement, networking, and regional partnerships. Few articles reported on whether and how long-term sustainability was ensured. Ideally, local researchers will develop and sustain programs in their settings. When those from HICs develop programs, interventions must prioritize sustainability with local LMIC leaders

spearheading the efforts after project completion [56]. Imbalanced partnerships will unlikely lead to serious sustainable capacity development in the South [57]. By LMIC researchers leading projects after completion, power imbalances can be shifted to promote sustainability. Several of these findings are in line with a commentary by a group of African researchers to concentrate on addressing research gaps that are relevant to policies and programs, carrying out high-quality and collaborative research, and translating research findings into policies and programs to achieve RCS through the synergistic commitment of global researchers, funders and organizations [58].

Gaps in the scoping review literature about how SRH research programs influence patient outcomes and health policy remain. The articles did not address gender diversity and did not mention built-in tracker systems. Even though the search strategy included non-English language articles, only articles written in English met the inclusion criteria. The English-language publications may reflect HIC-led programs' values and decision-making power [18]. According to MoChridhe, the high costs in time and money that using English in global scholarship imposes on non-native learners inhibits information exchange and impedes public participation in research, especially in regions with the greatest need for the opportunities that engagement with global scholarship promises and where articulate voices are most acutely missing from broader global conversations [59]. None of the papers in our sample specifically addressed training investigators on publishing their research. Publication of research by local investigators is critical to share findings for broader implementation and to inform policy [28, 60]. The gap in knowledge about SRH research programs in non-English speaking areas warrants further investigation.

Furthermore, while the work in Africa is commendable, greater geographic diversity in SRH research programs would benefit researchers living in other LMICs. Most studies were conducted in Africa, possibly due to the historical focus of funding entities. This Afrocentrism renders it difficult to generalize the collective findings to LMIC settings in other geographical regions [7]. Paradoxically, external and international funding remains critical to the sustainability of research and innovation systems in many African countries [61]. Four studies did not identify limitations. To ensure transparency and allow for adaption of programs, it is necessary to mention the limitations of interventions in articles.

Just as it is essential to identify the limitations of interventions, highlighting strengths and examples of what worked well in RCS programs can be beneficial. In an evaluation of the long-term impact of the Vanderbilt Institute in Research Development and Ethics, a decade-old intensive grant development practicum specifically tailored for investigators from LMICs, authors identified key program elements that contributed to the programs' success including a rigorous application and selection process led by LMIC partners, a curriculum based on applicable grant-writing skills, committed in-country and U.S. based mentorship teams, protected writing time, and an immersive cohort experience that provided participants with expanded professional networks [6]. These successful strategies could be translated to RCS specifically targeting SRH. Another example of what works well can be seen in the positive impact of blended courses on learners' work behavior and organizational achievements after participating in Primary Health Care (eSSP), Management of Sexual and Reproductive Health Services (eSSR), and Research Methods (eMR) training developed and implemented by Millimouno and colleagues with the Ministry of Health in Guinea [38]. Online or blended learning opportunities for SRH RCS could be a way to reach a broader global audience, provided internet connectivity is reliable.

In the review, several articles in our sample reported on interventions as they were in progress or shortly after completion. It would be helpful for more reports to assess post-intervention progress in SRH research after a year or more. Sustainability is critical to maintaining

SRH research program progress. A review of institutionalizing RCS in LMICs by Vicente-Crespo et al. [62] highlighted that three factors, equitable international partnerships, local leadership, and availability of funds, were related to greater intervention sustainability. Another way to foster the sustainability of RCS programs could be to institute an intensive mentorship programme combining hierarchical (vertical) and peer-to-peer (horizontal) mentoring strategies among young researchers, as undertaken by Balandya and colleagues in Tanzania [63]. In this RCS program, the less experienced peers received mentorship from senior researchers from a consortium of three partnering large Tanzanian health training institutions (MUHAS, CUHAS, and KCMUCo) and two collaborating US institutions (UCSF and Duke University) through mentored research awards and research training, and in turn provided reciprocal peer-to-peer mentorship as well as mentorship to undergraduate students [63].

Reported outcomes in our sample of scoping review studies were as varied as the methodology. In a review of RCS outcome indicators, Pulford and colleagues [64] found significant overlap and duplication in reported outcome indicators and identified priority focal areas, including research management and support, the attainment and application of new skills and knowledge, research collaboration, and knowledge transfer. When analyzing outcomes of SRH research programs, it is essential to keep in mind, due to differences in resources and research environments between LMICs and HICs, outcome measures commonly used in HICs, including publications in peer-reviewed journals, conference presentations, and funded grant applications, may not reflect accurately the research capacity and productivity in LMICs [6]. To move towards real inclusion in outcome indicators and global health in general, diverse voices, including the involvement of people who have historically been marginalized (i.e., women, those working in a low-income country, non-Anglophone, young people), need to be brought to the table to shape the debate and set the agenda [48, 65]. Furthermore, to move progress forward, research capacity outcomes need to be equally valued as research outputs [10], and health practitioners in HICs must strive to master the art of critical allyship and enablers rather than leading outcomes [56].

We conducted this scoping review to inform efforts to successfully implement current and future programs to strengthen SRH research programs while fostering long-term sustainability. One implication of the findings highlighted in the review is the need to foster the sustainability of SRH research programs with enduring collaborative partnerships in LMICs. Building and sustaining institutional support and individual engagement in SRH research takes time and effort. Further work is needed to incorporate long-term monitoring and evaluation into short-term training programs. From the onset, country ownership of a program needs to be emphasized along with the involvement of stakeholders, mainly the Ministry of Health, and leadership in local academic institutions.

Findings from the review are in line with the Guttmacher–Lancet Commission assertion that SRH are essential for the well-being and survival of individuals, economic prosperity, and the overall welfare of humanity [66]. Advancing SRH necessitates addressing the obstacles entrenched within legal frameworks, policies, economic structures, and societal norms and values, particularly gender disparities, which hinder individuals from attaining optimal sexual and reproductive health [66]. The legal framework and public policies hold equal significance alongside research in SRH [66]. Recent findings on the classification of abortion highlight that in countries with highly restrictive abortion laws, there is a notably higher prevalence of unsafe abortions compared to those with less stringent regulations. It is typical for LMICs to have both restrictive abortion laws and limited access to reproductive rights [67].

Across all included articles, SRH RCS programs diverged, making comparison between programs difficult. Acknowledging the power imbalances between LMICs and HICs and that the global research agenda is currently set and funded by HICs is important when reviewing

SRH research programs. Shifting the power to LMICs is essential because local researchers will define research priorities appropriate to their region, develop contextualized responses to local health problems, and connect research to policy and practice [56, 65]. To support sustainability, future RCS programs could consider having the leadership begin as a collaborative effort between LMIC and HIC team members and then transition to LMIC leadership.

## Study strengths

This is the first known review of RCS programs focused on SRH. The number of authors screening in this review allowed for multiple perspectives and reduced potential bias towards the opinion of fewer authors. A global health informationist, one of our team members guided the literature review process following PRISMA guidelines.

## Limitations

Given the wide contextual variety of included articles, it is difficult to compare across programs. The methodology of included articles is not directly transferable, yet the RCS programs included in the review can be adapted to local contexts. Due to the broad and diverse scope of the topic, although an effort was made to conduct a comprehensive search of the relevant literature, it is possible that articles meeting search criteria were missed.

## Conclusion

The results of this scoping review can be used to strengthen SRH research programs in LMICs. To address additional gaps, future research directions include exploring SRH research mentorship, identifying the cost of SRH RCS interventions, and how to foster institutional support for SRH research. More funds and energy must be directed toward ensuring sustainability in SRH research programs. To improve SRH in LMICs, we advocate for increasing SRH research strengthening programmatic support from funding organizations, institutions, and governments.

## Supporting information

**S1 Checklist. Preferred Reporting Items for Systematic reviews and Meta-Analyses extension for Scoping Reviews (PRISMA-ScR) checklist.**
(DOCX)

**S1 Appendix. Populated data extraction form that was used to gather information for the 18 articles included in this scoping review of reproductive health research strengthening programs in low- and middle-income countries.**
(DOCX)

## Author Contributions

**Conceptualization:** Julie M. Buser.

**Formal analysis:** Julie M. Buser, Anna Grace Auma, Ella August, Gurpreet K. Rana, Rachel Gray, Faelan E. Jacobson-Davies, Tamrat Endale, Yolanda R. Smith.

**Methodology:** Julie M. Buser, Gurpreet K. Rana.

**Software:** Gurpreet K. Rana.

**Supervision:** Yolanda R. Smith.

**Validation:** Julie M. Buser, Anna Grace Auma, Ella August, Gurpreet K. Rana.

**Visualization:** Faelan E. Jacobson-Davies.

**Writing – original draft:** Julie M. Buser, Anna Grace Auma, Ella August, Gurpreet K. Rana, Rachel Gray, Faelan E. Jacobson-Davies, Yolanda R. Smith.

**Writing – review & editing:** Julie M. Buser, Anna Grace Auma, Ella August, Gurpreet K. Rana, Rachel Gray, Faelan E. Jacobson-Davies, Tesfaye H. Tufa, Tamrat Endale, Madeleine Mukeshimana, Yolanda R. Smith.

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
