## [Decision Letter · Decision Letter 0]

12 Feb 2024

PGPH-D-23-02103

Reproductive health research capacity strengthening programs in low- and middle-income countries: A scoping review

Dear Dr. Buser,

Thank you for submitting your manuscript to PLOS Global Public Health. After careful consideration, we feel that it has merit but does not fully meet PLOS Global Public Health’s publication criteria as it currently stands. Therefore, we invite you to submit a revised version of the manuscript that addresses the points raised during the review process.

We look forward to receiving your revised manuscript.

Kind regards,

María De Jesús Medina Arellano, PhD

Academic Editor

Journal Requirements:

Additional Editor Comments (if provided):

Thank you so much for your paper, we just got the responses of both of our reviewers, please do go through all of the comments, so the paper could be reconsidered.

Reviewers' comments:

Reviewer's Responses to Questions

**Comments to the Author**

1. Does this manuscript meet PLOS Global Public Health’s publication criteria? Is the manuscript technically sound, and do the data support the conclusions? The manuscript must describe methodologically and ethically rigorous research with conclusions that are appropriately drawn based on the data presented.

Reviewer #1: Yes

Reviewer #2: Yes

2. Has the statistical analysis been performed appropriately and rigorously?

Reviewer #1: N/A

Reviewer #2: N/A

3. Have the authors made all data underlying the findings in their manuscript fully available (please refer to the Data Availability Statement at the start of the manuscript PDF file)?

Reviewer #1: Yes

Reviewer #2: Yes

4. Is the manuscript presented in an intelligible fashion and written in standard English?

Reviewer #1: Yes

Reviewer #2: Yes

5. Review Comments to the Author

Reviewer #1: The authors could consider these topics to improve some ideas:

1st. It is very common to treat the “sexual and reproductive” health or rights, when reproductive aspects are analyzed. Gender identity, gender expression, sexual orientation, etc. are no necessarily regularly considered, and their impact in health are very important. I think it is important to reconsider that in the tittle is written “Reproductive health research capacity”, a key word is “Reproductive health”, and in the introduction there is an ampliation to “Sexual and reproductive health”. It should be uniform.

2nd. There is a relatively old discussion regarding the relationship between research an public policies. The introduction says “There remains a limited understanding of the effectiveness of SRH research strengthening programs to raise skill sets, publications, and infrastructure and ultimately influence health policy and patient outcomes in LMICs.” Health research has its own logic, but also policies have another logic. Maybe, in an ideal world, policies should be guided by evidence (Ham C, Hunter DJ, Robinson R. Evidence based policymaking. BMJ. 1995 Jan 14;310(6972):71-2. doi: 10.1136/bmj.310.6972.71.) However, this has been considered at least as an incomplete vision of the problem (Azuonye IO. Evidence based policymaking. Academic experts don't have a monopoly on wisdom. BMJ. 1995 Apr 29;310(6987):1141. doi: 10.1136/bmj.310.6987.1141b.)

3rd. In the field of reproductive health (or sexual and reproductive health) this is known that “Sexual and reproductive health and rights (SRHR) are fundamental to people's health and survival, to economic development, and to the wellbeing of humanity.” (Starrs AM, et al. Accelerate progress-sexual and reproductive health and rights for all: report of the Guttmacher-Lancet Commission. Lancet. 2018 Jun 30;391(10140):2642-2692. doi: 10.1016/S0140-6736(18)30293-9.) Legal framework and public policies are as relevant as research in reproductive health.

4th. A clear example about previous topic is abortion. Recent research on abortion classification shows that “When grouped by the legal status of abortion, the proportion of unsafe abortions was significantly higher in countries with highly restrictive abortion laws than in those with less restrictive laws.” (Ganatra B, et al. Global, regional, and subregional classification of abortions by safety, 2010-14: estimates from a Bayesian hierarchical model. Lancet. 2017 Nov 25;390(10110):2372-2381. doi: 10.1016/S0140-6736(17)31794-4.) It is common that LMICs have restrictive abortion laws as well restrictive access to reproductive rights (or sexual and reproductive rights).

5th. Normative aspects should include not only legal ones, but ethical too. Authors say that “when participating in the research, they are often not given appropriate credit”. They could cite that “Guideline 8: Collaborative partnership and capacity-building for research and research review” (International Ethical Guidelines for Health-related Research Involving Humans (https://cioms.ch/wp-content/uploads/2017/01/WEB-CIOMS-EthicalGuidelines.pdf) which analyzes the problem and propose some solutions.

Reviewer #2: This is a potentially useful review.

The authors state that there is limited understanding of the effectiveness of SRH research strengthening programs to raise skill sets, publications, and infrastructure and ultimately influence health policy and patient outcomes in LMICs and that more information is needed to understand how SRH research is sustained after program completion. It is not clear how these stated gaps in literature are factored into the study. Is the goal of the review to describe literature on the effectiveness of SRH RCS or to describe the the format, duration, and topics covered of program interventions? A reader gets the feeling of a disconnect between the stated gaps and the study findings. The authors could focus merely on describing the interventions and the outcomes as described in the published literature. The study does not resolve/ answer the mentioned question of the effectiveness of RCS intervention. This means the paper has to be reintroduced to focus on the issues addressed in the review.

Answering the question of effectiveness would require a review of assessments/evaluation of SRH RCS focused-work.

Second, it will be great if the authors could clarify how they dealt with papers that focused on RCS in the field of public health generally. Some published studies on RCS interventions in public health include trainees who work on a diversity of health related topics of which SRH will be one. Did such paper qualify for inclusion? My point is that some RCS program include SRH but as a component of health generally.

Further questions include: how do such programs ensure gender diversity? Do they have built-in tracker systems; what are they key research skills focused on and not focused on etc.

The authors' point that "More energy must be directed toward correcting power imbalances in capacity strengthening initiatives" appears casually thrown in and does not reflect anything from the review.

Rethinking the paper with my first and second point above in mind will help improve the value of this manuscript.

6. PLOS authors have the option to publish the peer review history of their article (what does this mean?). If published, this will include your full peer review and any attached files.

**Do you want your identity to be public for this peer review?** For information about this choice, including consent withdrawal, please see our Privacy Policy.

Reviewer #1: **Yes: **Jorge Alberto Álvarez Díaz

Reviewer #2: No

---

## [Decision Letter · Decision Letter 1]

10 Jul 2024

PGPH-D-23-02103R1

Sexual and reproductive health research capacity strengthening programs in low- and middle-income countries: A scoping review

Dear Dr. Buser,

Thank you for submitting your manuscript to PLOS Global Public Health. After careful consideration, we feel that it has merit but does not fully meet PLOS Global Public Health’s publication criteria as it currently stands. Therefore, we invite you to submit a revised version of the manuscript that addresses the points raised during the review process.

EDITOR: Please insert comments here and delete this placeholder text when finished. Be sure to:

Indicate which changes you require for acceptance versus which changes you recommendAddress any conflicts between the reviews so that it's clear which advice the authors should followProvide specific feedback from your evaluation of the manuscript

Please ensure that your decision is justified on PLOS Global Public Health’s publication criteria and not, for example, on novelty or perceived impact.

We look forward to receiving your revised manuscript.

Kind regards,

María De Jesús Medina Arellano, PhD

Academic Editor

Journal Requirements:

Additional Editor Comments (if provided):

Please, find attached the revisions made and try to clarify and modify accordingly in order to resubmit your manuscript. ThNAK YOU.

Reviewers' comments:

Reviewer's Responses to Questions

**Comments to the Author**

1. If the authors have adequately addressed your comments raised in a previous round of review and you feel that this manuscript is now acceptable for publication, you may indicate that here to bypass the “Comments to the Author” section, enter your conflict of interest statement in the “Confidential to Editor” section, and submit your "Accept" recommendation.

Reviewer #1: All comments have been addressed

Reviewer #3: (No Response)

2. Does this manuscript meet PLOS Global Public Health’s publication criteria? Is the manuscript technically sound, and do the data support the conclusions? The manuscript must describe methodologically and ethically rigorous research with conclusions that are appropriately drawn based on the data presented.

Reviewer #1: Yes

Reviewer #3: Yes

3. Has the statistical analysis been performed appropriately and rigorously?

Reviewer #1: N/A

Reviewer #3: N/A

4. Have the authors made all data underlying the findings in their manuscript fully available (please refer to the Data Availability Statement at the start of the manuscript PDF file)?

Reviewer #1: Yes

Reviewer #3: Yes

5. Is the manuscript presented in an intelligible fashion and written in standard English?

Reviewer #1: Yes

Reviewer #3: Yes

6. Review Comments to the Author

Reviewer #1: I consider the requirements were fully fulfill.

Reviewer #3: The authors were responsive to the previous reviews, and I just have a couple of minor suggestions to further improve the manuscript. First, I noticed that the revision from "reproductive health" to "sexual and reproductive health" was in response to previous reviews, and I agree with the revision. However, the introduction still seems focused on reproductive health or RH, and it gets a little confusing. Please double-check for consistent terminology throughout.

Second, it would be helpful to explain and justify why the timeframe of 2011-2023 was chosen.

7. PLOS authors have the option to publish the peer review history of their article (what does this mean?). If published, this will include your full peer review and any attached files.

**Do you want your identity to be public for this peer review?** For information about this choice, including consent withdrawal, please see our Privacy Policy.

Reviewer #1: **Yes: **Jorge Alberto Álvarez Díaz

Reviewer #3: No

---

## [Decision Letter · Decision Letter 2]

13 Sep 2024

Sexual and reproductive health research capacity strengthening programs in low- and middle-income countries: A scoping review

PGPH-D-23-02103R2

Dear Dr. Buser,

We are pleased to inform you that your manuscript 'Sexual and reproductive health research capacity strengthening programs in low- and middle-income countries: A scoping review' has been provisionally accepted for publication in PLOS Global Public Health.

Best regards,

María De Jesús Medina Arellano, PhD

Academic Editor

Reviewer Comments (if any, and for reference):

Reviewer's Responses to Questions

**Comments to the Author**

1. If the authors have adequately addressed your comments raised in a previous round of review and you feel that this manuscript is now acceptable for publication, you may indicate that here to bypass the “Comments to the Author” section, enter your conflict of interest statement in the “Confidential to Editor” section, and submit your "Accept" recommendation.

Reviewer #1: All comments have been addressed

Reviewer #3: All comments have been addressed

2. Does this manuscript meet PLOS Global Public Health’s publication criteria? Is the manuscript technically sound, and do the data support the conclusions? The manuscript must describe methodologically and ethically rigorous research with conclusions that are appropriately drawn based on the data presented.

Reviewer #1: Yes

Reviewer #3: Yes

3. Has the statistical analysis been performed appropriately and rigorously?

Reviewer #1: N/A

Reviewer #3: N/A

4. Have the authors made all data underlying the findings in their manuscript fully available (please refer to the Data Availability Statement at the start of the manuscript PDF file)?

Reviewer #1: Yes

Reviewer #3: (No Response)

5. Is the manuscript presented in an intelligible fashion and written in standard English?

Reviewer #1: Yes

Reviewer #3: (No Response)

6. Review Comments to the Author

Reviewer #1: I think authors have included all the recommendations and improved the paper. I think it could be accepted.

Reviewer #3: (No Response)

7. PLOS authors have the option to publish the peer review history of their article (what does this mean?). If published, this will include your full peer review and any attached files.

**Do you want your identity to be public for this peer review?** For information about this choice, including consent withdrawal, please see our Privacy Policy.

Reviewer #1: **Yes: **Jorge Alberto Álvarez Díaz

Reviewer #3: No
